# The Effect of Starting Blood Glucose Levels on Serum Electrolyte Concentrations during and after Exercise in Type 1 Diabetes

**DOI:** 10.3390/ijerph20032109

**Published:** 2023-01-24

**Authors:** Zeinab Momeni, Normand G. Boulé, Carla M. Prado, Heather A. Hinz, Jane E. Yardley

**Affiliations:** 1Augustana Faculty, University of Alberta, 4901-46th Avenue, Camrose, AB T4V 2R3, Canada; 2Physical Activity and Diabetes Laboratory, Alberta Diabetes Institute, 112 Street, Edmonton, AB T6G 2T9, Canada; 3Women’s and Children’s Health Research Institute, University of Alberta, Edmonton, AB T6G 1C9, Canada; 4Faculty of Kinesiology, Sport, and Recreation, University of Alberta, 3-100 University Hall, Van Vliet Complex, Edmonton, AB T6G 2H9, Canada; 5Human Nutrition Research Unit, Alberta Diabetes Institute, 112 Street, Edmonton, AB T6G 2T9, Canada; 6Faculty of Agricultural, Life and Environmental Science, University of Alberta, Edmonton, AB T6G 2P5, Canada

**Keywords:** blood glucose, exercise, type 1 diabetes, electrolytes, hydration

## Abstract

Fear of hypoglycemia is a major exercise barrier for people with type 1 diabetes (PWT1D). Consequently, although guidelines recommend starting exercise with blood glucose (BG) concentration at 7–10 mmol/L, PWT1D often start higher, potentially affecting hydration and serum electrolyte concentrations. To test this, we examined serum and urine electrolyte concentrations during aerobic exercise (cycling 45 min at 60%VO_2_peak) in 12 PWT1D (10F/2M, mean ± SEM: age 29 ± 2.3 years, VO_2_peak 37.9 ± 2.2 mL·kg^−1^·min^−1^) with starting BG levels: 8–10 (MOD), and 12–14 (HI) mmol/L. Age, sex, and fitness-matched controls without diabetes (CON) completed one exercise session with BG in the normal physiological range. Serum glucose was significantly higher during exercise and recovery in HI versus MOD (*p* = 0.0002 and *p* < 0.0001, respectively) and in MOD versus CON (*p* < 0.0001). During exercise and recovery, MOD and HI were not significantly different in serum insulin (*p* = 0.59 and *p* = 0.63), sodium (*p* = 0.058 and *p* = 0.08), potassium (*p* = 0.17 and *p* = 0.16), calcium (*p* = 0.75 and 0.19), and magnesium *p* = 0.24 and *p* = 0.09). Our findings suggest that exercise of moderate intensity and duration with higher BG levels may not pose an immediate risk to hydration or serum electrolyte concentrations for PWT1D.

## 1. Introduction

In people with type 1 diabetes (PWT1D), being physically active is associated with increased longevity [1] along with a decrease in the risk and/or progression of diabetes-related complications [2,3]. Unfortunately, fear of hypoglycemia acts as a major barrier to being physically active in this population [4]. Those who are active often sacrifice tight glycemic management and start their activities with higher than recommended blood glucose (BG) levels as a means to avoid hypoglycemia [5]. In a survey of 502 PWT1D, participants reported a mean pre-exercise BG target of 7.8 ± 1.5 mmol/L, with the upper limit of the range being 14.4 mmol/L [5]. In addition, only 50% of respondents reported that they would delay starting exercise if BG levels were high, with an average cut-off of 11.7 ± 3.4 mmol/L, but a range that stretched up to 16.7 mmol/L [5]. A smaller survey (n = 45) of athletes with T1D found that almost 25% of participants reported having BG levels in excess of 10.0 mmol/L both at the start, and at the end of their training sessions [6].

In the absence of physical activity/exercise, once BG levels exceed approximately 9–10 mmol/L, the tubular maximum for glucose reabsorption in the kidney is reached [7], leading to glucose in the urine and a loss of water by osmotic diuresis [8]. Hyperglycemia can also increase sodium reabsorption by both the distal [9] and proximal [10] tubules in the kidney, as well as increase serum potassium levels [11], and urinary magnesium excretion. Taken together, these factors may interfere with muscle contraction and body cooling (especially sweating), which may partially explain why PWT1D experience elevated morbidity and mortality during heat waves compared to their counterparts without diabetes [12]. Whether or not these imbalances are exacerbated by exercise among PWT1D has yet to be examined.

Two small observational studies involving athletes with T1D measured renin and aldosterone but did not examine changes in electrolytes and hydration during exercise [13,14]. These measurements were also taken while BG levels were between 7 and 9 mmol/L. To the best of our knowledge, changes in hydration and serum electrolyte concentrations during exercise have not been examined in PWT1D, neither in a euglycemic, nor a hyperglycemic state. Therefore, the present study aimed to examine hydration and serum electrolyte concentrations during a standardized exercise session in physically active PWT1D starting with BG close to the currently recommended target zone (8–10 mmol/L) [15] and a slightly higher target (12–14 mmol/L) in comparison to age, height, weight, and fitness-matched individuals without diabetes. We hypothesized that exercise performed with higher BG levels would lead to greater water loss, higher serum sodium and potassium concentration, and lower serum magnesium concentration in PWT1D than when the same exercise is performed with lower BG levels. We also hypothesized that when exercise is performed closer to a euglycemic range, changes in hydration and serum electrolyte concentrations would be similar to matched control participants without diabetes.

## 2. Materials and Methods

Habitually active PWT1D aged between 18 and 50 years were recruited to take part in the study. Snowball sampling and convenience sampling (i.e., family members, colleagues, friends, etc.) were used to find matched controls without diabetes. Exclusion criteria included a T1D diagnosis of less than 2 years, HbA1c greater than 9.9%, frequent and/or unpredictable hypoglycemia or hypoglycemia unawareness, change in diabetes management strategy in the previous two months, use of medication (other than insulin) that alters glucose metabolism, or any condition/complication that would contraindicate exercise (i.e., uncontrolled hypertension, clinically significant gastroparesis, proliferative retinopathy, lower limb injury). Participants were matched for sex, with the goal of also matching closely for physical fitness (VO_2_peak), age (±5 years), weight (±2 kg), and height (±10 cm). All volunteers provided written informed consent. The study was approved by the University of Alberta Health Research Board (Biomedical) in accordance with the Declaration of Helsinki.

### 2.1. Baseline Testing and Measurement of VO_2_peak

Participants attended a baseline session to establish consent and eligibility. During the session, height, weight, and seated blood pressure measurements were taken. Fitness was determined using a peak oxygen consumption (VO_2_peak) test, during which participants performed an incremental exercise test to volitional exhaustion on a Monark Ergomedic 894E Peak Bike (Monark, Varberg, Sweden). Heart rate was recorded using a Polar heart rate monitor watch and belt (Polar Electro Oy, Kempele, Finland). Oxygen consumption and carbon dioxide production were measured using a Parvo Medics TrueOne^®^ Metabolic Measurement System (Sandy, UT, USA).

### 2.2. Experimental Design

Participants with T1D undertook two testing sessions, one with starting BG levels between 8 and 10 mmol/L (MOD) and one with starting BG levels between 12 and 14 mmol/L (HI), while control participants performed a single session with BG in their normal physiological range (CON). The order of the MOD and HI testing sessions was pre-determined by flipping a fair coin during the baseline visit. MOD and HI tests were separated by at least 48 h. Participants with T1D were asked to make whatever adjustments they deemed necessary to reach the target BG level before the start of the exercise. If necessary, carbohydrates in the form of Dex4 tablets (AMG Medical, Montreal, QC, Canada) were provided to reach the target prior to exercise. If needed, carbohydrates were also provided throughout the exercise in order to maintain BG levels within the desired range.

All tests took place at the same time of day (arrival at the lab for ~16 h for a ~17 h exercise start). At least 24 h before the first testing session, participants visited the lab to have a Medtronic Enlite^®^ CGM sensor inserted subcutaneously in their abdominal region and attached to a blinded iPro2^®^ recorder (Medtronic, Minneapolis, MN, USA). The CGM remained in place until at least 24 h after the final testing session. All participants recorded food intake and insulin dosage in a food log and replicated their food intake (timing and composition) for the meals on the day before, day of, and day after CON, MOD, and HI testing sessions. They also recorded at least four daily capillary glucose values for CGM calibration purposes. The consecutive three-day food intake record during CGM monitoring days was analyzed for macronutrient content using the ESHA Food Processor Nutritional Analysis software (version 11.7, ESHA Research Inc., Salem, OR, USA) based on the Canadian Nutrient File. Background physical activity was monitored using a pedometer (Yamax Digiwalker 200, Yamax Corporation, Tokyo, Japan), with steps being recorded in the food log.

### 2.3. Exercise Testing Sessions

Exercise testing sessions consisted of a 45 min bout of moderate-intensity (60% VO_2_peak) aerobic exercise performed on a cycle ergometer, followed by one hour of seated recovery. Five-minute indirect calorimetry measurements were performed after 5 min and 35 min of exercise to confirm participant oxygen consumption levels and adjust the resistance as necessary. The rating of perceived exertion and heart rate were also recorded every 5 min during exercise to monitor the intensity of the session. Participants did not consume any water during exercise. At the end of the 45 min exercise period, they were provided with a 500 mL bottle of water with half of the volume to be consumed within the first 10 min of recovery and the remainder to be consumed before the 30 min mark of the recovery period. It was felt that this design best replicated the water consumption patterns of most habitually active individuals [6].

### 2.4. Blood and Urine Sample Collection and Analysis

To facilitate blood sample collection throughout exercise and recovery, an IV catheter was inserted upon arrival at the laboratory. Venous blood samples were taken immediately before exercise (time 0), at the end of exercise (45 min), and at the end of recovery (105 min). Blood samples were taken in a seated position on an orthodontic-like chair with the seat having a 45-degree angle. To ensure participant safety, and to maintain BG in the desired range during exercise, capillary glucose was measured at least every 10 min using a hand-held glucometer (OneTouch Ultra2, LifeScan, Milpitas, CA, USA). This measurement was done due to a lag time with CGM during aerobic exercise in adults with T1D, which raises the need to measure capillary BG in a setting where the risk of hypoglycemia is elevated [16]. Venous blood samples were collected and were immediately mixed by inversion. The K_2_EDTA BD Vacutainer^®^ tubes were centrifuged immediately for the separation of plasma, while the BD Vacutainer^®^ serum separator tubes (SST) were kept at room temperature for at least 15 min before centrifugation at 3000 revolutions/minute for 15 min. Plasma and serum samples were then aliquoted and stored at −80 °C for subsequent analysis. Frozen serum aliquots were shipped on dry ice to DynaLIFE Medical Labs (Edmonton, AB, Canada) for the analysis of glucose (using glucose hexokinase), sodium (using ion-selective electrode), potassium (using ion-selective electrode), calcium (using o-cresolphthalein complexone without deproteinization), magnesium (using modified xylidyl blue reaction), and insulin (using chemiluminescent microparticle immunoassay). It should be noted that the insulin analogs have cross-reactivity (binding of the antibodies employed in the assay to determine the concentration of an analyte) with reagents used for insulin measurement in this study (ARCHITECT^®^ insulin assay) [17]. In particular, the ARCHITECT insulin assay has shown high cross-reactivity to the insulin analogs lispro and glargine, and low cross-reactivity to the insulin analog aspart [17]. Our study participants used NovoRapid and Fiasp (insulin aspart), Humalog (insulin lispro), Toujeo (insulin glargine), Tresiba, or a combination of these analogs.

Participants were also provided with a sterile container in order to provide urine samples at the same time points, i.e., immediately before exercise, at the end of exercise, and at the end of recovery. Urine specific gravity (U_SG_) was assessed immediately using a clinical refractometer (Reichert Inc., NY, USA). Urine samples were aliquoted and frozen at −80 °C for subsequent batch analysis. Similarly, frozen urine aliquots were shipped on dry ice to DynaLIFE Medical Labs (Edmonton, AB, Canada) for the analysis of sodium, potassium, and calcium. Urine glucose levels were measured using the Cayman glucose colorimetric assay kit (Cayman Chemical, MI, USA).

### 2.5. CGM Data Assessment

CGM data were examined in the following windows: 0–6 h post-exercise, overnight (00:00 to 06:00), and 24 h post-exercise. Coefficient of variation (CV: standard deviation/mean) and standard deviation (SD) of CGM glucose were calculated for each timepoint as measures of variability. The percent of time spent in euglycemia (3.9–10 mmol/L), level 1 hypoglycemia (3.0–3.8 mmol/L), level 2 hypoglycemia (<3.0 mmol/L), level 1 hyperglycemia (10.1–13.9 mmol/L), and level 2 hyperglycemia (>13.9 mmol/L), as well as the frequency of level 1 and level 2 hypo- and hyperglycemia were also calculated. All CGM data were assessed in line with the current recommendations on CGM reporting and interpretation [18,19].

### 2.6. Statistical Analysis

For all statistical tests, the comparisons were performed separately for MOD versus HI (to examine the first hypothesis, i.e., the impact of hyperglycemia on serum electrolyte concentrations and hydration within the context of T1D, using repeated measures) and CON versus MOD (to examine the second hypothesis, i.e., the impact of T1D on serum electrolyte concentrations and hydration when BG is close to the physiological range). GraphPad Prism 9 (San Diego, CA, USA) was used for all the statistical tests performed. Two-way ANOVA, with Bonferroni post-hoc test, was used to examine the main and interaction effects between treatments (i.e., CON vs. MOD, and MOD vs. HI, separately) and time (i.e., pre- vs. post-exercise, and post-exercise vs. recovery, separately) for serum glucose, insulin, sodium, potassium, calcium, and magnesium, as well as the urine glucose, sodium, potassium, calcium, and U_SG_. Based on the normality of the data, paired or unpaired t-test, Mann–Whitney U test, or Wilcoxon signed-rank test were used to compare CGM data within each time point. Two-way ANOVA, with Bonferroni post-hoc test, was also used to examine the main and interaction effects between treatments (i.e., CON vs. MOD, and MOD vs. HI) and time (day 1: the day before vs. day 2: day of vs. day 3: the day after exercise) for daily food intake, total daily insulin, and background physical activity (steps per day). A *p*-value of < 0.05 was considered significant for all statistical tests performed.

## 3. Results

Twelve (2 male and 10 female) PWT1D (3 using multiple daily insulin injections, 9 using continuous subcutaneous insulin infusion, 4 using their own flash/continuous glucose monitoring) and 12 age, fitness, height, weight, and sex-matched control participants completed the protocol at the University of Alberta between January 2018 and April 2020 (Table 1). There was no significant difference in terms of the percentage of individual VO_2_peak for exercise intensity when MOD (62 ± 3.4) was compared to HI (62.3 ± 3.8) or when CON (62.1 ± 3.4) was compared to MOD during the testing sessions (*p* > 0.05). There was no significant difference between MOD (12 ± 0.9) and HI (12.4 ± 1.1) in the rating of perceived exertion (RPE—Borg 6 to 20-point scale) during the testing sessions (*p* > 0.05), while there was a significant difference between CON (13.2 ± 1.5) and MOD testing sessions (*p* = 0.02).

### 3.1. MOD vs. HI Comparisons for the Main Outcomes

Values for all serum and urinary measures can be found in Appendix A. Serum glucose decreased from the beginning to the end of exercise during both MOD and HI sessions (time effect: *p* = 0.001) and increased during recovery (time effect: *p* = 0.0002), with serum glucose being significantly lower in MOD versus HI during both exercise (treatment effect: *p* = 0.0002, interaction: *p* = 0.51) and recovery (treatment effect: *p* = 0.0104, interaction: *p* = 0.9) (Figure 1A).

Serum insulin did not change significantly from the beginning to the end of exercise or during recovery (time effect: *p* = 0.57, and *p* = 0.75, respectively) during both MOD and HI sessions, nor was it significantly different between the two sessions during exercise (treatment effect: *p* = 0.59, interaction: *p* = 0.09) or recovery (treatment effect: *p* = 0.63, interaction: *p* = 0.31) (Figure 1B).

Serum sodium concentration increased significantly from the beginning to the end of exercise in both MOD and HI sessions (time effect: *p* = 0.003) and decreased during recovery (time effect: *p* < 0.0001), with a nonsignificant trend towards higher serum sodium in MOD versus HI during both exercise (treatment effect: *p* = 0.058, interaction: *p* = 0.62) and recovery (treatment effect: *p* = 0.08, interaction: *p* = 0.56) (Figure 1C).

Serum potassium and calcium concentrations increased from the beginning to the end of exercise in both MOD and HI sessions (time effect: *p* < 0.05) and decreased during recovery (time effect: *p* < 0.05). Serum magnesium concentration showed an increasing trend from the beginning to the end of exercise during MOD and HI sessions that did not reach statistical significance (time effect: *p* = 0.057). It, however, decreased significantly during the recovery (time effect: *p* < 0.0001) during both MOD and HI sessions. Serum concentrations of potassium, calcium, and magnesium were not significantly different between MOD and HI sessions during exercise or recovery (treatment effect: *p* > 0.05, interaction: *p* > 0.05) (Figure 1D–F).

Urine glucose concentration did not change significantly from the beginning to the end of exercise (*p* = 0.35) or during recovery (*p* = 0.93) during the MOD session, while it increased significantly from the beginning to the end of exercise (*p* = 0.01) but not during the recovery (*p* = 0.23) during the HI session. Urine glucose concentration was significantly lower in MOD during both exercise (treatment effect: *p* = 0.002, interaction: *p* = 0.006) and recovery (treatment effect: *p* = 0.0001, interaction: *p* = 0.19) when compared to HI (Figure 2A).

Urine sodium, potassium, and calcium concentrations, and U_SG_ did not change significantly from the beginning to the end of exercise or during the recovery (time effect: *p* > 0.05) in MOD and HI sessions, nor were they significantly different when MOD was compared to HI during exercise or recovery (treatment effect: *p* > 0.05, interaction: *p* > 0.05) (Figure 2B–E).

Mean CGM glucose and measures of glucose variability were not significantly different between MOD and HI sessions during any of the time points (*p* > 0.05) (Table 2).

### 3.2. CON vs. MOD Comparisons for the Main Outcomes

Serum glucose did not change significantly during exercise (*p* > 0.9) or recovery (*p* > 0.9) in CON, but it was significantly lower during both exercise (treatment effect: *p* < 0.0001, interaction: *p* = 0.002) and recovery (treatment effect: *p* < 0.0001, interaction: *p* < 0.0001) when compared to MOD. Significant interaction also suggests that the way in which serum glucose changed during exercise and recovery was significantly different between the two groups (Figure 3A).

Serum insulin decreased from the beginning to the end of exercise (*p* = 0.02) in CON, with serum insulin being significantly lower than in MOD during both exercise (treatment effect: *p* = 0.02, interaction: *p* = 0.01) and recovery (treatment effect: *p* = 0.01, interaction: *p* = 0.23) (Figure 3B).

Serum sodium concentration did not change significantly during exercise (*p* > 0.9) or recovery (*p* = 0.9) in CON, nor was it significantly different from MOD during exercise (treatment effect: *p* = 0.56, interaction: *p* = 0.02) or recovery (treatment effect: *p* = 0.9, interaction: *p* = 0.01). Significant interaction, however, suggests that while serum sodium concentration was not different between CON and MOD, the way in which it changed during exercise and recovery was significantly different between the two groups (Figure 3C).

Serum concentrations of potassium, calcium, and magnesium increased from the beginning to the end of exercise (time effect: *p* < 0.05) and decreased during recovery (time effect: *p* < 0.05) in CON. Serum potassium concentration was not significantly different between CON and MOD during exercise (treatment effect: *p* = 0.33, interaction: *p* = 0.5) or recovery (treatment effect: *p* = 0.9, interaction: *p* = 0.7). Serum calcium concentration was also not significantly different between CON and MOD during exercise (treatment effect: *p* = 0.65, interaction: *p* = 0.2), but it was significantly higher in CON during the recovery (treatment effect: *p* = 0.03, interaction: *p* = 0.29). Serum magnesium concentration was significantly higher during both exercise (treatment effect: *p* = 0.0002, interaction: *p* = 0.22) and recovery (treatment effect: *p* < 0.0001, interaction: *p* = 0.18) in CON versus MOD (Figure 3D–F).

Urine glucose concentration and U_SG_ did not change significantly during exercise (time effect: *p* > 0.05) or recovery (time effect: *p* > 0.05) in CON, but they were significantly lower during both exercise (treatment effect: *p* < 0.05, interaction: *p* > 0.05) and recovery (treatment effect: *p* < 0.05, interaction: *p* > 0.05) when compared to MOD (Figure 4A,B).

Urine sodium and calcium concentrations were not significantly different when CON was compared to MOD during exercise or recovery (treatment effect: *p* > 0.05, interaction: *p* > 0.05), while urine potassium concentration was significantly lower in MOD during both exercise (treatment effect: *p* = 0.008, interaction: *p* = 0.12) and recovery (treatment effect: *p* = 0.003, interaction: *p* = 0.6). During CON, urine potassium and calcium concentrations did not change significantly during exercise or recovery (time effect: *p* > 0.05), while urine sodium concentration decreased from the beginning to the end of exercise (*p* = 0.03) but did not change significantly during the recovery (*p* = 0.06) (Figure 4C–E).

Mean CGM glucose, SD, CV, time in level 1 hyperglycemia, and frequency of level 1 hyperglycemia were all significantly lower in CON when compared to MOD during all time points (*p* < 0.05). Additionally, time in level 1 hypoglycemia, frequency of level 1 hypoglycemia, and frequency of level 2 hyperglycemia were significantly lower in CON versus MOD during the 24 h post-exercise (*p* < 0.05). Percent of time spent in range was significantly higher in CON versus MOD during all time points (*p* < 0.05) (Table 2).

### 3.3. Food and Insulin Intake and Background Physical Activity

Food diary analysis showed no significant difference in the mean fat, carbohydrate, protein, or total energy intake when three days were compared within each group (Table 3). The mean carbohydrate intake, however, was significantly higher on both day 2 and day 3 in CON compared to MOD. Total daily insulin dose showed no significant difference between three days within MOD or HI, or when MOD was compared to HI. Background physical activity was also similar on all three monitored days and in all three groups (Table 3).

## 4. Discussion

In this study, we examined serum and urine electrolyte concentrations during aerobic exercise in participants with T1D having two different starting BG levels, i.e., MOD and HI, as well as in matched controls. Contrary to our hypothesis, we did not find any significant differences between MOD and HI sessions in serum or urine electrolyte concentrations or mean CGM glucose, hypo/hyperglycemia, and measures of glucose variability in the 24 h post-exercise. Also contrary to our hypothesis, there were significant differences in serum and urine electrolyte concentrations, mean CGM glucose, and measures of glucose variability between CON and MOD, i.e., even when the exercise was performed closer to a euglycemic range.

As expected, our data showed decreasing serum glucose levels, but no change in serum insulin levels in T1D participants during exercise in both MOD and HI sessions. Meanwhile, there were no significant changes in serum glucose levels, but a decrease in serum insulin levels from the beginning to the end of exercise in CON participants. These differences in the change in insulin during exercise are reflective of the different behaviors of endogenous and synthetic insulin.

Our data also showed a significantly lower concentration of urine glucose in CON versus MOD and in MOD versus HI during both exercise and recovery. As BG levels rise, renal tubular reabsorption of glucose increases until it reaches its maximum tubular resorptive capacity (renal threshold—usually around 9–10 mmol/L), [20] after which glucose is excreted in the urine [7]. Thus, it can be speculated that HI participants excreted more glucose during both exercise and recovery than MOD (and MOD versus CON) due to having higher BG levels at the start and throughout the exercise session. Measurement of urinary glucose excretion is, however, required to confirm this. There is currently a lack of evidence to determine whether glucosuria is increased in the context of exercise in PWT1D.

Urine specific gravity was within the normal physiological range (1.003–1.030) [21] during both exercise and recovery in CON, MOD, and HI. Although U_SG_ was not significantly different between MOD and HI sessions, it was significantly lower in CON when compared to MOD during both exercise and recovery. This higher U_SG_ during MOD may be caused by renal glucosuria [22] due to having a higher BG level. Furthermore, U_SG_ correlates with urine osmolality and reflects hydration status [23], with a value of greater than 1.020 indicating relative dehydration [24]. During MOD, U_SG_ was slightly above 1.020 at the end of exercise and recovery (1.0207 ± 0.002 and 1.0206 ± 0.002, respectively), which could indicate relative dehydration as compared to other sessions in which U_SG_ was below 1.020, indicating euhydration in all time points. Due to significant knowledge gaps in regard to U_SG_ during exercise with different BG levels, we cannot fully explain at this point why the HI session was not associated with a higher U_SG_ despite having higher glucosuria.

Serum sodium concentration was lower in CON when compared to the MOD session. As sodium-glucose co-transporter 2 (SGLT2) in the proximal tubule mediates the reabsorption of glucose and sodium [25], it can be inferred that higher BG levels during the MOD session might have led to increased sodium reabsorption in this group as compared to CON. These trends are consistent with previous studies that reported elevated sodium reabsorption in both proximal [10] and distal [9] tubules of the kidney in response to high BG levels. Comparison between MOD and HI, on the other hand, did not show a significant difference in serum sodium concentration between the two sessions, which might be partly due to higher glucosuria in HI, where the renal reabsorption threshold is exceeded.

Serum potassium concentration was not significantly different between CON and MOD or between MOD and HI sessions. These findings are consistent with a small study that found similar plasma potassium concentrations between T1D and control participants both at rest and in response to a single supra-maximal sprint to exhaustion [26]. The increase in potassium throughout exercise in CON, MOD, and HI is consistent with an increase in the release of potassium from the intracellular to extracellular space of skeletal muscles, following repetitive action potentials, and further into the bloodstream in response to exercise as reported elsewhere [27].

A possible mechanism explaining the lower urine potassium concentration in CON versus MOD during exercise and recovery could be increased activation of the renal outer medullary potassium channels in MOD due to lower magnesium concentration (as observed in the serum magnesium results) [28,29]. Lower plasma magnesium concentration has been reported in PWT1D compared to people without diabetes [30,31], which may be attributed to higher levels of urinary excretion of magnesium in those with T1D compared to non-diabetic controls [32].

Serum calcium concentration was significantly higher in CON when compared to MOD during the recovery. Although diabetes-induced hypocalcemia may occur as a result of renal insufficiency and accompanying hyperphosphatemia [29], this does not seem to be the case in our T1D participants since serum calcium concentration is still within the normal physiological range in all three time points. Diabetes-associated impairment of intestinal calcium absorption, as well as uncoupling of bone remodeling, i.e., suppression of osteoblast function and enhancement of osteoclastogenesis and osteoclast function, are other suggested mechanisms for impaired calcium metabolism in those with diabetes [33]. Further investigation, however, is still required to determine why the reduction in serum calcium concentration happened during recovery but not during the exercise in MOD participants as compared to the CON.

Serum magnesium concentration increased during exercise in CON and decreased significantly during the recovery in CON, MOD, and HI. The results of acute and chronic effects of exercise on serum and plasma concentration of magnesium are highly variable [34], mostly due to the type, intensity, and duration of the exercise. Sixty minutes of ergometer bicycling at approximately 65% of the estimated maximum capacity did not change plasma magnesium significantly during exercise but decreased it during the first hour of recovery in males without T1D [35]. Although this study was closest to the present study in terms of the mode and intensity and is consistent with the reduction we observed during recovery, different results achieved during the exercise in CON may be due to sex differences in fluid and electrolyte balance during activity [36] since the majority of our participants were female. It should be noted that the concentrations of serum sodium (135–145 mmol/L), potassium (3.6–5.2 mmol/L), calcium (2.1–2.6 mmol/L), and magnesium (0.7–1 mmol/L) were all within the normal physiological range before the start of exercise, at the end of exercise, and at the end of a 60 min recovery in CON, MOD, and HI.

Mean CGM glucose and measures of glucose variability showed significant differences between CON and MOD, as expected, but not between MOD and HI. The lack of significant differences between MOD and HI shows that being hyperglycemic during exercise neither increases nor decreases the risk of post-exercise hypoglycemia in PWT1D. These findings are consistent with a secondary analysis of a moderately large dataset (120 participants) pooled from previously published studies where higher pre-exercise glucose concentration was associated with a greater drop in BG during 60 min of moderate aerobic exercise, albeit with less risk of post-exercise hypoglycemia [37]. These glycemic trends can be explained by differences in fuel metabolism when aerobic exercise is performed in a euglycemic versus a hyperglycemic state in PWT1D, with a greater reliance on carbohydrates as a fuel source during exercise when BG levels are high [38].

Although the CGM data presented in this study were all collected with the iPro2, we did not ask those who were already using their own/unblinded CGM (n = 4) as part of their regular management to discontinue their use. This can be considered a limitation since it could potentially affect glucose management and/or time in hypo/hyperglycemia. We also noticed dietary under-reporting by the majority of our participants, making it difficult to assess the impact of dietary intake. Furthermore, although participants were asked to arrive well-hydrated with their BG levels in a target range, the means by which they achieved their starting BG levels were not standardized. This may have increased the variability in some measured outcomes. In addition, the baseline urine sample contained urine that accumulated over variable periods of time, probably within the two hours before exercise, i.e., when preparation for exercise started in participants. Finally, we do not have a record of urinary volume for each timepoint, and urinary concentrations are largely dependent on the urinary volumes produced. We were not able to measure urine electrolyte balance in our study as it could not be estimated without measures of urinary electrolyte excretion and urinary volumes. Further research is required to determine the total solute excretion in each participant for each time point to better address urine electrolyte balance in response to exercise with different starting BG levels.

## 5. Conclusions

Although exercise was performed closer to a euglycemic range during the MOD session, there were significant differences in serum and urine electrolyte concentrations as well as CGM outcomes between CON and MOD sessions. Furthermore, contrary to our hypothesis, no significant differences were found between MOD and HI sessions in serum or urine electrolyte concentrations or mean CGM outcomes. This lack of significant differences between MOD and HI sessions can suggest that exercise with higher BG levels may not pose an immediate health risk during the exercise of moderate intensity and duration in otherwise healthy adults with type 1 diabetes. However, we cannot rule out the possibility that a longer, or potentially more intense, exercise session at a different temperature could alter the impact of hyperglycemia on serum and urine electrolyte concentrations. Further research is therefore warranted.

## Figures and Tables

**Figure 1 ijerph-20-02109-f001:**
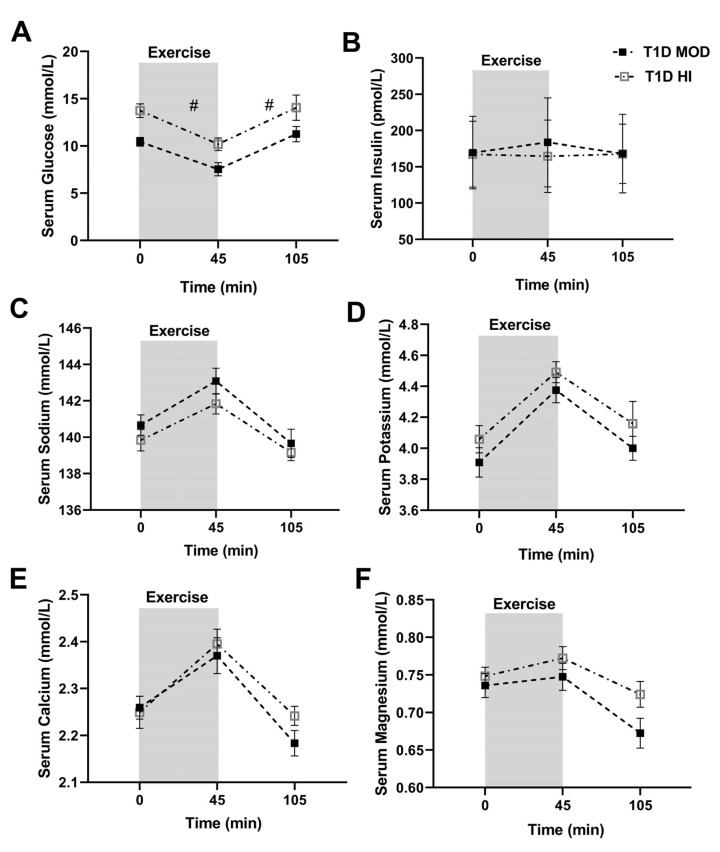
Serum concentrations (mean ± SEM) of glucose (**A**), insulin (**B**), sodium (**C**), potassium (**D**), calcium (**E**), and magnesium (**F**) before exercise (time 0), at the end of exercise (45 min), and at the end of the 60 min recovery (105 min) in T1D MOD (black squares) and T1D HI (grey squares). # shows significant differences (treatment effect) between MOD and HI from the beginning to the end of exercise and from the end of exercise to the end of recovery compared by ANOVA.

**Figure 2 ijerph-20-02109-f002:**
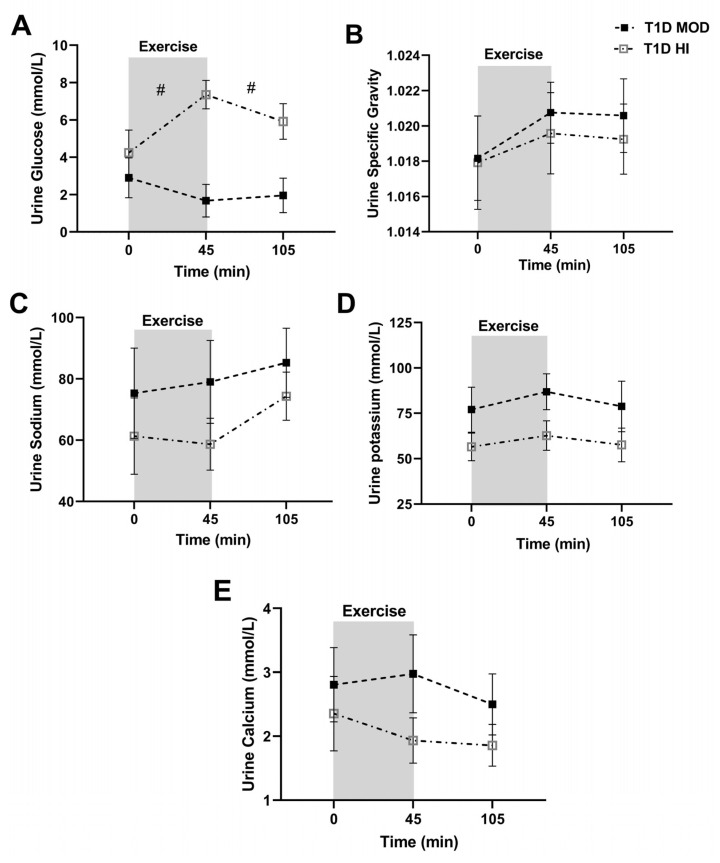
Urine concentration (mean ± SEM) of glucose (**A**), urine specific gravity (**B**), and urine concentrations of sodium (**C**), potassium (**D**), and calcium (**E**) before exercise (time 0), at the end of exercise (45 min), and at the end of the 60-min recovery (105 min) in T1D MOD (black squares) and T1D HI (grey squares). # shows significant differences (treatment effect) between MOD and HI from the beginning to the end of exercise and from the end of exercise to the end of recovery compared by ANOVA.

**Figure 3 ijerph-20-02109-f003:**
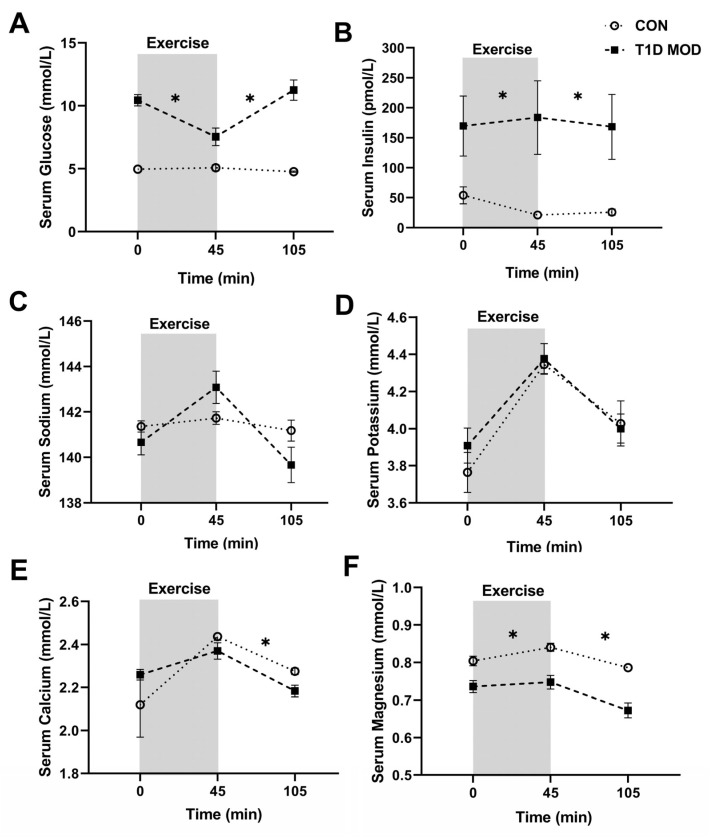
Serum concentrations (mean ± SEM) of glucose (**A**), insulin (**B**), sodium (**C**), potassium (**D**), calcium (**E**), and magnesium (**F**) before exercise (time 0), at the end of exercise (45 min), and at the end of the 60 min recovery (105 min) in CON (white circles) and T1D MOD (black squares). * shows significant differences (treatment effect) between CON and MOD from the beginning to the end of exercise and from the end of exercise to the end of recovery, compared by ANOVA.

**Figure 4 ijerph-20-02109-f004:**
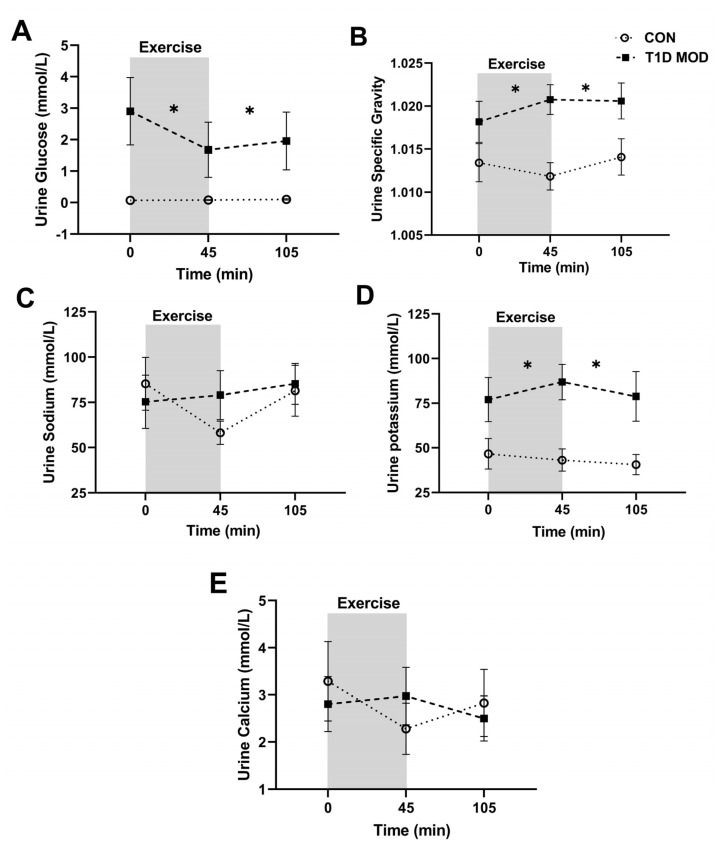
Urine concentration (mean ± SEM) of glucose (**A**), urine specific gravity (**B**), and urine concentrations of sodium (**C**), potassium (**D**), and calcium (**E**) before exercise (time 0), at the end of exercise (45 min), and at the end of the 60 min recovery (105 min) in CON (white circles) and T1D MOD (black squares). * shows significant differences (treatment effect) between CON and MOD from the beginning to the end of exercise and from the end of exercise to the end of recovery compared by ANOVA.

**Table 1 ijerph-20-02109-t001:** Participant characteristics presented as mean ± SD.

Characteristic	T1D Participants	Control Participants	*p*-Value
M/F (n)	2/10	2/10	n/a
Age (yrs)	29.0 ± 7.8	29.3 ± 7.9	0.93
Height (cm)	165.3 ± 7.3	169.1 ± 8.0	0.24
Weight (kg)	69.4 ± 7.9	68.7 ± 8.4	0.82
BMI (kg/m^2^)	25.4 ± 2.0	24 ± 2.5	0.16
VO_2_peak (mL·kg^−1^·min^−1^)	37.9 ± 7.7	38.4 ± 7.7	0.86

**Table 2 ijerph-20-02109-t002:** Post-exercise glucose and glycemic variability in control and PWT1D measured by CGM.

		CON (n = 12)	T1D MOD (n = 12)	T1D HI (n = 12)	*p* (CON vs. MOD)	*p* (MOD vs. HI)
0–6 h post-exercise	Mean CGM glucose (mmol/L)	5.7 (5.2–6.2)	8.7 (6.8–10.1)	9.8 (8.4–11.5)	<0.0001 *	0.08
	SD (mmol/L)	0.5 (0.3–0.6)	1.8 (0.9–2.6)	2.1 (1.3–2.8)	<0.0001 *	0.51
	CV (%)	8.8 (5.9–11.1)	21.6 (13.3–28.6)	22.2 (13.2–25.7)	0.0002 *	0.85
	Time in range (%)	100 (100–100)	70.3 (59.0–95.4)	52.2 (30.2–84.0)	<0.0001 *	0.07
	Time in level 1 hypoglycemia (%)	0.0 (0.0–0.0)	0.2 (0.0–0.0)	2.0 (0.0–0.0)	>0.99	0.5
	Time in level 2 hypoglycemia (%)	0.0 (0.0–0.0)	0.0 (0.0–0.0)	0.7 (0.0–0.0)	n/a	0.5
	Time in level 1 hyperglycemia (%)	0.0 (0.0–0.0)	24.4 (4.5–39.6)	36.7 (16.0–63.5)	<0.0001 *	0.18
	Time in level 2 hyperglycemia (%)	0.0 (0.0–0.0)	4.9 (0.0–5.2)	8.3 (0.0–18.3)	0.09	0.29
	Frequency of level 1 hypoglycemia	0.0 (0.0–0.0)	0.0 (0.0–0.0)	0.16 (0.0–0.0)	n/a	0.5
	Frequency of level 2 hypoglycemia	0.0 (0.0–0.0)	0.0 (0.0–0.0)	0.08 (0.0–0.0)	n/a	>0.99
	Frequency of level 1 hyperglycemia	0.0 (0.0–0.0)	1.1 (0.2–2)	1.4 (1–2)	0.0003 *	0.44
	Frequency of level 2 hyperglycemia	0.0 (0.0–0.0)	0.3 (0.0–1)	0.5 (0–1)	0.09	0.68
Overnight (00:00 to 06:00)	Mean CGM glucose (mmol/L)	5.5 (5.1–5.9)	8.7 (7.0–9.9)	8 (7.2–9.0)	<0.0001 *	0.44
	SD (mmol/L)	0.27 (0.15–0.29)	1.7 (0.8–2.7)	1.6 (1.3–2.1)	0.0001 *	0.82
	CV (%)	5.0 (2.8–5.2)	19.2 (11.6–27.1)	20.6 (16.0–24.9)	<0.0001 *	0.48
	Time in range (%)	100 (100–100)	66.3 (30.6–100.0)	65.8 (43.0–85.7)	0.001 *	0.86
	Time in level 1 hypoglycemia (%)	0.0 (0.0–0.0)	2.3 (0.0–0.0)	0.46 (0.0–0.0)	0.21	0.34
	Time in level 2 hypoglycemia (%)	0.0 (0.0–0.0)	8.8 (0.0–0.0)	7.3 (0.0–0.0)	0.47	0.73
	Time in level 1 hyperglycemia (%)	0.0 (0.0–0.0)	17.4 (0.0–26.4)	21.5 (7.6–32.6)	0.004 *	0.43
	Time in level 2 hyperglycemia (%)	0.0 (0.0–0.0)	5.1 (0.0–0.0)	4.86 (0.0–3.1)	0.21	0.82
	Frequency of level 1 hypoglycemia	0.0 (0.0–0.0)	0.2 (0.0–0.0)	0.08 (0.0–0.0)	0.21	0.59
	Frequency of level 2 hypoglycemia	0.0 (0.0–0.0)	0.0 (0.0–0.0)	0.08 (0.0–0.0)	n/a	>0.99
	Frequency of level 1 hyperglycemia	0.0 (0.0–0.0)	0.7 (0.0–1)	0.9 (1–1)	0.001 *	0.40
	Frequency of level 2 hyperglycemia	0.0 (0.0–0.0)	0.1 (0.0–0.0)	0.2 (0.0–0.7)	0.47	0.59
0–24 h post-exercise	Mean CGM glucose (mmol/L)	5.6 (5.2–5.8)	7.8 (6.7–8.8)	8.7 (7.4–9.9)	0.002 *	0.29
	SD (mmol/L)	0.5 (0.46–0.53)	2.2 (1.2–3.1)	2.5 (1.6–3.2)	<0.0001 *	0.5
	CV (%)	9.3 (8.4–10.0)	29.6 (19.3–39.0)	29 (20.1–36.7)	<0.0001 *	0.84
	Time in range (%)	100 (100–100)	73.3 (61.5–94.2)	63.8 (49.0–81.3)	<0.0001 *	0.31
	Time in level 1 hypoglycemia (%)	0.0 (0.0–0.0)	1.7 (0.0–3.4)	2.3 (0.0–2.8)	0.004 *	0.68
	Time in level 2 hypoglycemia (%)	0.0 (0.0–0.0)	0.35 (0.0–0.0)	2.4 (0.0–0.3)	>0.99	0.32
	Time in level 1 hyperglycemia (%)	0.0 (0.0–0.0)	21.5 (4.2–29.5)	23.2 (10.4–31.8)	<0.0001 *	0.49
	Time in level 2 hyperglycemia (%)	0.0 (0.0–0.0)	3 (0.0–7.1)	8.3 (0.0–16.3)	0.03 *	0.24
	Frequency of level 1 hypoglycemia	0.0 (0.0–0.0)	0.5 (0.0–1.0)	0.5 (0.0–1.0)	0.013 *	0.75
	Frequency of level 2 hypoglycemia	0.0 (0.0–0.0)	0.08 (0.0–0.0)	0.2 (0.0–0.0)	>0.99	0.59
	Frequency of level 1 hyperglycemia	0.0 (0.0–0.0)	2.5 (1.2–3.0)	3.0 (2.0–4.0)	<0.0001 *	0.24
	Frequency of level 2 hyperglycemia	0.0 (0.0–0.0)	0.0 (0.0-1.0)	1.3 (0.0–3.0)	0.03 *	0.24

As recommended by the international consensus on CGM reporting, data are reported as median [IQR]. CGM: continuous glucose monitoring, CV: coefficient of variation, SD: standard deviation. Level 1 hypoglycemia is defined as CGM glucose between 3.0 and 3.8 mmol/L. Level 2 hypoglycemia is defined as CGM glucose < 3.0 mmol/L. Time in range is defined as CGM glucose between 3.9 and 10 mmol/L. Level 1 hyperglycemia is defined as CGM glucose between 10.1 and 13.9 mmol/L. Level 2 hyperglycemia is defined as CGM glucose > 13.9 mmol/L. *p*-values within each timepoint result from the paired or unpaired t-test, Mann–Whitney U test, or Wilcoxon signed-rank test based on normality of the data. N/A refers to when two groups had zero difference and comparison between the two was not applicable. * *p* < 0.05.

**Table 3 ijerph-20-02109-t003:** Participants’ daily fat, carbohydrate, and protein intakes; daily energy intake; total daily insulin; and steps per day presented as mean ± SEM.

	CON (n = 12)			T1D MOD (n = 12)			T11D HI (n = 11)		
	Day 1	Day 2	Day 3	Day 1	Day 2	Day 3	Day 1	Day 2	Day 3
Fat intake (g)	54.1 ± 8.8	66.6 ± 8.9	55.5 ± 7.5	57.6 ± 13.3	50.0 ± 4.3 *	44.9 ± 6.4	52.5 ± 8.8	53.9 ± 5.2	47.3 ± 8.2
Carbohydrate intake (g)	157.6 ± 19.1	200.7 ± 24.1	187.2 ± 20.7	120.6 ± 19.2	127.8 ± 15.4 *	102.2 ± 17.8 *	112.8 ± 17.1	140.3 ± 18.5	141.8 ± 28.3
Protein intake (g)	68.4 ± 11.1	85.7 ± 12	72.2 ± 10.4	67.4 ± 14.3	83.9 ± 13.7	64.7 ± 15.7	63.3 ± 14.6	87.5 ± 14.1	73.4 ± 7.4
Daily energy intake (kcal)	1368 ± 179	1717 ± 204	1521 ± 162	1240 ± 208	1274 ± 110	1069 ± 156	1165 ± 176	1377 ± 107	1248 ± 170
Total daily insulin (U)	n/a	n/a	n/a	29.9 ± 6.9 (n = 11)	28.9 ± 4.9 (n = 11)	25.8 ± 3.1 (n = 11)	24.9 ± 4.3	25.5 ± 4.5	25.5 ± 4.3
Steps per day	5176 ± 1071	7099 ± 1134	6317 ± 1251	6633 ± 1157	8623 ± 1301	7421 ± 1198	6289 ± 659	9581 ± 931	7648 ± 1604

* significant difference (*p* < 0.05) between CON and MOD when compared for the same day.

## Data Availability

Most of the data generated and/or analyzed during the current study are included in this article. Other data will be available from the corresponding author upon reasonable request.

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
