# Peer review of "The Effect of Starting Blood Glucose Levels on Serum Electrolyte Concentrations during and after Exercise in Type 1 Diabetes"

_ijerph, 2023, doi:10.3390/ijerph20032109_

Round 1

Reviewer 1 Report (Previous Reviewer 1)

Thank you for this re-submission.  I have provided a more detailed review in the previous submission.  While I disagree with the statistical approach used, I feel that the revisions made are sufficient to convey this meaningful research.  I will respect the authors presentation decision. The figure presentation is now much more clear in relation to your stated hypotheses.  The only thing I might suggest is to present the results of CON vs MOD before MOD vs HIGH in the results.  This isn't a big deal, but may help the data flow in a logical (in my mind) order.  Again, thanks for your thoughtful revisions.

Author Response

We thank the reviewer for again taking the time to review our work. While we understand that the reviewer might find the comparison between CON and MOD more interesting than that of MOD and HIGH, this comparison was not the main objective of the study. We feel that it would be inappropriate to present our secondary outcomes before our primary outcomes.  As such, we have made no further changes to the manuscript.

Reviewer 2 Report (New Reviewer)

The manuscript entitled "The Effect of Starting Blood Glucose Levels on Serum Electrolyte Concentrations During and After Exercise in Type 1 Diabetes" reports the results of a study aimed to assess the relationship between blood glucose levels and serum electrolytes changes during exercise in people with T1DM. This topic is of interest as it can represent a barrier to exercise and could be associated with a risk for health issues.

The study was well-designed, and the manuscript has been well-written by the authors; the different colors in the manuscript give the impression that this manuscript has been already reviewed and improved after previous considerations. Indeed, the text is clear.
I have no further comments or suggestions and would like to congratulate the authors for the interesting work.

Author Response

Thank you for taking the time to read our manuscript again.

This manuscript is a resubmission of an earlier submission. The following is a list of the peer review reports and author responses from that submission.

Round 1

Reviewer 2 Report

Overall, the paper is well written and researched. The population can be difficult to recruit and matching of control groups was done very well. However, the absence of many vital measures, urinary volumes, body mass changes during exercise is surprising and limits the value of the study findings. I may be in error but to my reading, it appears that the study was not specifically designed to address the aim of the study, but rather that these measures were made as part of a larger study.

Introduction

Line 57 The paper refers to other potential mechanisms e.g. autonomic neuropathy and glucose tolerance, so the word ‘partially explain’ may be better.

Methods

How was posture or duration of seating controlled during the study? This is unclear with the exception of the 1h of seated recovery post exercise. Posture and duration in the posture (e.g. duration of seated) affects fluid shifts e.g. plasma volume changes from sitting to standing to lying down. Postural changes could explain some fluid shifts e.g. if the pre-exercise sample was taken within 5 min of sitting down it would differ from those measures at 15 min as body fluid equilibrates.

Good matching or participants for age, sex, fitness.

Line 103 – the 48 h separation between trials is rather short, particularly when dehydration is likely to occur. Could this be justified in the paper to a greater extent.

Line 134 – cite evidence to support this

Results

Were post exercise body mass changes determined ie a measure of likely sweat losses during exercise? This could inform likely fluid shifts and electrolyte losses and also whether weight loss during exercise was the same in each condition.

Figure 2 A – it would be better to offset the axes rather than scale down to an impossible value (-5 mmol/l), since it is somewhat misleading. Similarly, Fig 2B would maximise visualization of differences in the axes was set at 1.010 minimum.

Line 238 and Figure 2 – Better to use the term ‘urine glucose concentration’ or ‘urine sodium concentration’ to avoid ambiguity as, for example, urine sodium could refer to either urine sodium concentration (mmol/L) or urine sodium excretion (mmol), which are two different things.

Unlike the other tables and figures, Table 2 is poorly formatted and difficult to read in the manuscript provided. Is all the data necessary to present as it makes it difficult for the reader to interpret given it’s length and the lack of significant differences between conditions?

Table 3 – clear evidence of under-reporting in each group given the energy (better to use energy intake rather than caloric intake). This data is not referred to in the Discussion, but although dietary under-reporting is very common, it is remarkably low in the present study (e.g. mean intake Day 3 MOD = 1069 ± 156 kcal) raises some issues with the accuracy of the self-report data.

Discussion

The study is very interesting and valuable. My main concern is the urinary data, given that these urinary concentrations are largely dependent on urine volumes produced. In Line 443 this is referred to, but I would argue that the value of the urinary data is very limited given the absence of record of urine volumes. This major limitation needs to be addressed to a greater extent as it yields much of the urinary data not very useful. Urinary electrolyte balance cannot be estimated without measures of urinary electrolyte excretion and the absence of urinary volumes means that this is not possible with the dataset presented. The title of the article is ‘The Effect of Starting Blood Glucose Levels on Serum and Urine Electrolyte Balance During and After Exercise in Type 1 Diabetes’, but electrolyte balance was not measured and could not be measured in the absence of urinary volumes and body mass losses during exercise. Body fluid concentrations are not the same as body fluid balances.

Line 350 – urinary glucose excretion wasn’t measured, though you can speculate that this is the case.

Line 368 – was fluid intake prior to arrival in the lab not controlled? Based on Figure 1C, serum sodium (a hydration status marker) was lowest in HI (non-sig) in those with T1D, which would not support that they were more dehydrated from not drinking as much prior to exercise on HI.

Line 444 -As detailed earlier, this is one of the main fundamental issues with the study and can explain the lack of clarity of terms used in the manuscript with respect to whether the urinary measures are concentrations or quantities.

Thus, in my opinion, there are fundamental errors in the urinary measures used and this limits the usefulness of the study findings. A greater focus on the blood measures would improve the manuscript, but I have some concerns about whether factors influencing the blood born measures such as posture are sufficiently controlled for.

Round 2

Reviewer 1 Report

Thank you for your thoughtful edits.  Many things were address to make this manuscript better.  I previously commented on the appropriateness of the statistical approach used.  I understand the authors response and could agree with this approach (although, technically not correct).  However, the rest of the manuscript presentation does not follow this approach.  At the minimum the figures should reflect the study design.  

My suggestion to the authors would be to think about two separate manuscripts.  My reasoning for this is 1) this would better reflect your study designs, 2) As stated in your response you were not interested in several of the comparisons, and 3) In your response you indicate that it would hard to include the applicable data in the abstract and that you felt that several important points may detract the focus.  Perhaps separate papers would help keep the focus.